# Characterization of the Environmental Plasmidome of the Red Sea

Lucy Androsiuk,[a,b,c] Tal Shay,[c] (iD) Shay Tal[a]

[a]Israel Oceanographic & Limnological Research Ltd., National Center for Mariculture, Eilat, Israel
[b]Marine Biology and Biotechnology Program, Department of Life Sciences, Ben-Gurion University of the Negev, Eilat, Israel
[c]Department of Life Sciences, Ben-Gurion University of the Negev, Beer-Sheva, Israel

**ABSTRACT**  Plasmids contribute to microbial diversity and adaptation, providing microorganisms with the ability to thrive in a wide range of conditions in extreme environments. However, while the number of marine microbiome studies is constantly increasing, very little is known about marine plasmids, and they are very poorly represented in public databases. To extend the repertoire of environmental marine plasmids, we established a pipeline for the *de novo* assembly of plasmids in the marine environment by analyzing available microbiome metagenomic sequencing data. By applying the pipeline to data from the Red Sea, we identified 362 plasmid candidates. We showed that the distribution of plasmids corresponds to environmental conditions, particularly, depth, temperature, and physical location. At least 7 of the 362 candidates are most probably real plasmids, based on a functional analysis of their open reading frames (ORFs). Only one of the seven has been described previously. Three plasmids were identified in other public marine metagenomic data from different locations all over the world; these plasmids contained different cassettes of functional genes at each location. Analysis of antibiotic and metal resistance genes revealed that the same positions that were enriched with genes encoding resistance to antibiotics were also enriched with resistance to metals, suggesting that plasmids contribute site-dependent phenotypic modules to their ecological niches. Finally, half of the ORFs (50.8%) could not be assigned to a function, emphasizing the untapped potential of the unique marine plasmids to provide proteins with multiple novel functions.

**IMPORTANCE**  Marine plasmids are understudied and hence underrepresented in databases. Plasmid functional annotation and characterization is complicated but, if successful, may provide a pool of novel genes and unknown functions. Newly discovered plasmids and their functional repertoire are potentially valuable tools for predicting the dissemination of antimicrobial resistance, providing vectors for molecular cloning and an understanding of plasmid-bacterial interactions in various environments.

**KEYWORDS**  marine plasmids, plasmidome, antibiotic resistance, metal resistance, Red Sea

Plasmids are extrachromosomal, mainly circular, double-stranded DNA (dsDNA) elements found in many species of *Bacteria*, *Archaea*, and *Eukarya* (1). Typically, plasmids harbor replication control, partitioning, and mobilization genes, ensuring their transfer and maintenance within the cell. Some bacterial plasmids are mobile (2), thereby facilitating an exchange of genetic material between species and hence contributing to inter- and intra-species genomic diversity. Mobile plasmids may carry antibiotic and biocide resistance genes and genes coding for virulence factors (3) and will confer those characteristics on their new host. These accessory genes of plasmids are selected by the evolutionary forces in the particular ecological niches in which the host resides, and therefore, they constitute a key element in studies of microbial habitats and of the ability of microbes to survive under specific conditions (4–6).

Address correspondence to Shay Tal, shay.tal@ocean.org.il.

The authors declare no conflict of interest.

In the past, studies on plasmids were performed by culturing strains of host bacteria in bacterial collections (7). As a result, studies on plasmids derived from natural environments (8–11) were limited to plasmids from cultivatable bacteria, plasmids that can conjugate to cultivatable bacteria, and plasmids for which there was a known primer (12). The consequent lack of studies related to the ecology of plasmids in natural environments can be attributed not only to difficulties in culturing plasmid hosts (13–15) but also to challenges in isolating plasmid DNA (4, 6, 7, 16) and to the lack of tools to detect and quantify plasmids (16, 17). In recent years, however, marked progress has been made in addressing these problems and challenges as a result of the availability of relatively low-cost high-throughput sequencing technologies, which has increased the possibility of obtaining higher-quality data on plasmids. Thus, new studies have identified plasmids in metagenomic data sets from groundwater (13), bovine rumen (5, 18), wastewater (19), and marine (20, 21) samples. In all these studies, plasmids were enriched and amplified during the DNA extraction process to ensure that sufficient reads were sequenced to enable accurate *de novo* plasmid reconstruction in the analysis step.

In recent years, there has thus been a rapid increase in the number of known plasmids, from 13,789 bacterial plasmids recorded in the plasmid database PLSDB when it was first published (September 2018) to 34,513 records in the current version (June 2021; PLSDB v. 2021_06_23) (22), i.e., a 2.5-fold increase within less than 3 years. However, marine plasmids—constituting only approximately 1% of plasmids in the database—are clearly underrepresented. Thus, there are striking knowledge gaps pertaining to the distribution, genetic repertoire, and environmental impact of marine plasmids (23). In contrast, there has been a rapid accumulation of environmental microbiome studies in general, and specifically marine microbiome studies, with studies ranging from the limited 16S amplicon sequencing to full metagenome deep sequencing. However, most of these studies have focused on bacterial genomes, with no attempts to specifically identify plasmids, and hence they did not include specific plasmid amplification steps. Nonetheless, full metagenomic data may contain sufficient reads of plasmid DNA to provide information about plasmids. The deeper the sequencing, the greater the likelihood that the data will include sufficient plasmid reads to facilitate accurate *de novo* plasmid assembly (6, 13, 18, 24).

There are several tools for the detection of plasmids in metagenomics data (25–27), but most of these tools are based on alignment to known plasmids, which are mainly terrestrial and freshwater plasmids. The implication of the underrepresentation in the PLSDB (22) of marine plasmids, which are most probably different from terrestrial and freshwater plasmids (11, 28, 29), is that the majority of currently available methods are very limited for analyzing marine environmental data. Nevertheless, there are a few tools that are suitable for *de novo* assembly of marine plasmids from metagenomics data (17, 30, 31). While the accuracy of these *de novo* assemblers needs to be improved (31), they have indeed facilitated the discovery of novel plasmids in environments for which knowledge about plasmids is limited, such as marine environments.

In the current study, we constructed a pipeline for *de novo* plasmid assembly and detection from multiple related environmental samples and applied the pipeline to public metagenomics samples collected from the Red Sea (32) (see the full description in Materials and Methods). Of 362 plasmid candidates, we classified seven as probable plasmids (where only one of the seven has previously been reported). We showed a strong correlation between the plasmid distribution patterns in marine environments and the physical conditions of those environments (such as depth and temperature), while only some plasmid distribution patterns in marine environments correlated with microbial distribution patterns. The complex correlation structure between the physical conditions and the distribution patterns of plasmids and microorganisms illustrates the potential contribution of plasmids to the adaptation of bacteria to their ecological niches.

## RESULTS

**Construction of the marine plasmidome.** We established a pipeline (Fig. 1A) for plasmid identification and detection from raw short sequencing reads in metagenomic data sets that had not been enriched for plasmids. The pipeline uses both SPAdes (33) followed

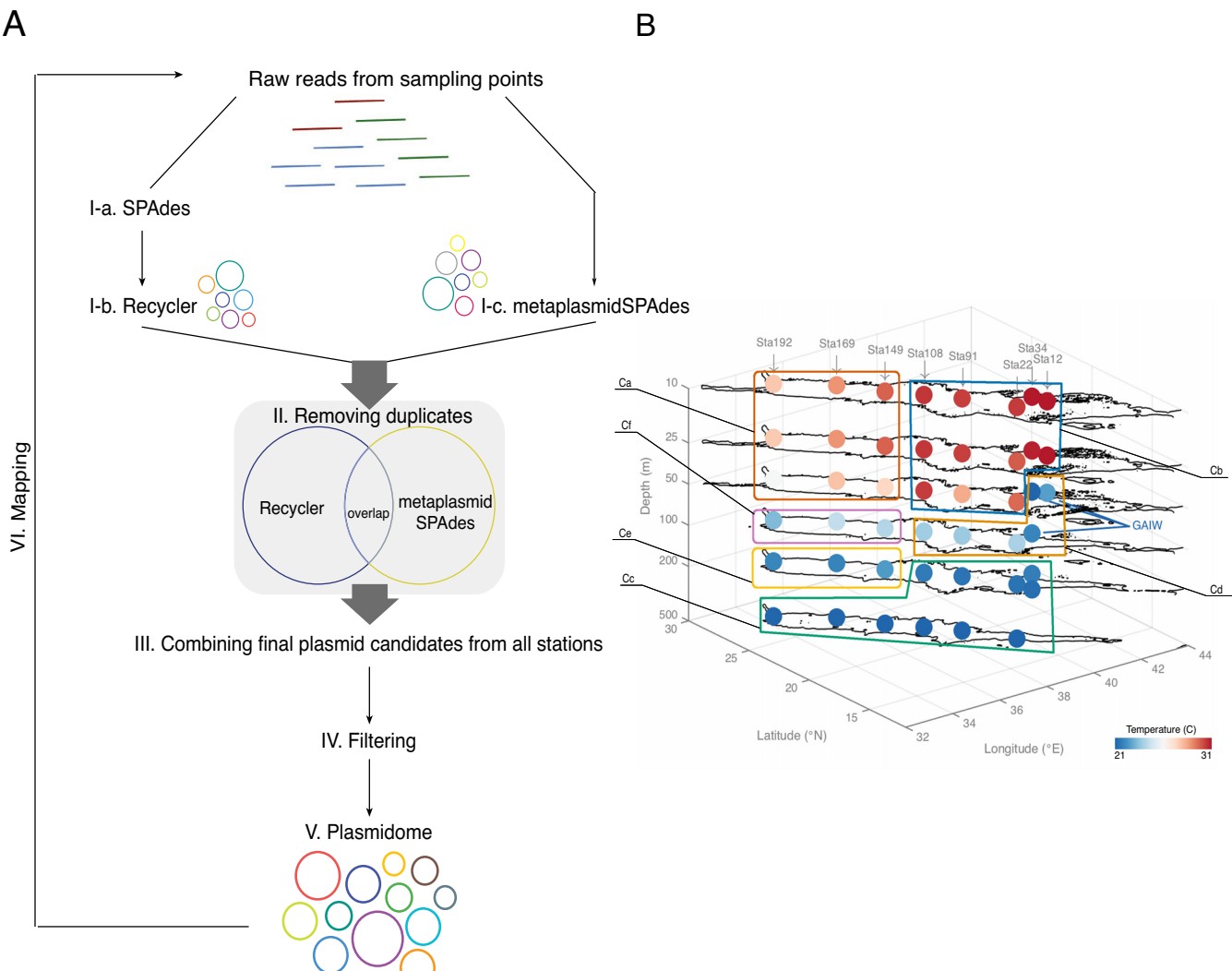

**FIG 1** Plasmid detection pipeline and data set. (A) Analysis overview. A metagenomic assembly graph was created for the raw reads of each sampling point by using SPAdes (33) (step I-a) followed by Recycler (17) (step I-b) or metaplasmidSPAdes (30) (step I-c). The resulting plasmid contigs were compared, and duplicates were removed (step II). Contigs from all stations were combined (step III) and filtered (step IV). Predicted plasmids were clustered according to sequence similarity, and one candidate from each cluster was selected as representative in the plasmidome (step V). Then, raw reads in each sampling point were mapped to the plasmidome to estimate presence of each plasmid at each station (step VI). (B) Three-dimensional map of sampling points (circles) in the Red Sea (figure adapted from the work of Haroon et al. [32]). Colors represent water temperature from low (dark blue) to high (dark red). Clusters Ca to Cf correspond to the sampling point clusters in Fig. 2A.

by Recycler (17) and metaplasmidSPAdes (30) for *de novo* plasmid assembly. We applied this pipeline to a public marine data set (32) composed of samples collected from eight stations along a cruise track in the Red Sea from south to north (14 to 26°N; 34 to 40°E) and from the surface to mesopelagic depths (10, 25, 50, 100, 200, and 500 m below sea level), as shown in Fig. 1B. Exceptions were the stations numbered 12 and 34, situated where the Gulf of Aden Intermediate Water (GAIW) enters the Red Sea (Fig. 1B): at station 12, samples were collected at depths of 10, 25, and 47 m, and at station 34, collection was at depths of 10, 25, 50, 100, 200, and 258 m. From a total of 45 sampling points, 576 plasmid candidates were predicted by Recycler, and 268 plasmid candidates were predicted by metaplasmidSPAdes. After filtering out duplicate plasmids, 362 unique candidates (average GC content = 42.66%) remained in the plasmidome (see files 1 and 2 in the supplemental material). Several events of partial candidate alignment were identified, which may indicate that some candidates share genes or that smaller candidates are part of larger ones.

**Plasmid presence patterns correlate with sampling point characteristics.** Although 260 of the 362 plasmid candidates (71.82%) were identified by Recycler and/or metaplasmidSPAdes at only one sampling point, it is possible that those plasmid candidates

are also present at other locations but were missed by the identification algorithm for various reasons; for example, low-level plasmids may yield a sequencing coverage that is too low to allow assembly and identification by our pipeline. Therefore, for each candidate, we calculated the percentage of candidate length covered by the raw reads at each sampling point (Fig. 2A). This additional targeted alignment step reduced the number of plasmids present (>99% of length covered) at only one sampling point from 260 to 34 of the 362 candidate plasmids (9.39%).

The next step was to perform hierarchal clustering of the sampling points according to plasmid presence patterns. The sampling point clusters Ca to Cf shown in Fig. 2A correspond to the physical conditions of the locations. Clustering the presence patterns of the plasmids produced eight clusters, designated C1 to C8. As expected, there were strong correlations between the different physical parameters; for example, the deeper the water, the lower the temperature and the lower the oxygen level (34). To investigate whether the plasmid patterns correspond to the conditions, we correlated the patterns of the eight clusters with the physical conditions of the locations (Fig. 2B). The correlation matrix revealed a strong positive correlation of clusters C5, C7, and C8 with the geographic location (latitude; Pearson correlation $P$ value $< 0.005$), whereas clusters C1 to C3 were strongly correlated with physical conditions, such as temperature and oxygen, nitrate, phosphate, and silicate concentrations (Pearson correlation $P$ value $< 0.005$).

**Correlation between plasmid and microbial distribution patterns.** As plasmids are not free-living entities and require a microbial host, we studied the association between the plasmid candidates and the microbial population. First, we reanalyzed the microbial distribution based on the analysis in reference 34, including all the microbial genera, and clustered the microbial population according to their distribution in the different sampling positions (Fig. 3A). As expected, the microbial distribution patterns also clustered according to the physical conditions in the different sampling positions; however, the clusters are less distinct than the clusters of plasmid candidates (Fig. 3D), indicating a higher degree of mixing at the microbial level. However, some plasmid candidates' distribution patterns are highly positively correlated with microbial distribution patterns (Fig. 3B; file 3 in the supplemental material). Interestingly, only 37.6% of the plasmid candidates strongly correlate (Pearson correlation coefficient $> 0.8$) with at least one microbial genus (Fig. 3C). For example, plasmid candidates' cluster C1 strongly correlates with microbial cluster Cb7, having a Pearson correlation coefficient of 0.97. Both clusters represent a deep-water population, mostly found in sampling cluster Cc (Fig. 3A and D), with microbial cluster Cb7 showing relatively high representation of archaeal microbes (27.3%), as expected in a deep-water environment.

**Marine plasmids are mostly novel.** Only six of the 362 (1.65%) plasmid candidates that we identified are also in the PLSDB (22), with coverage of >50% and identity percentage of >70% (Table 1; a full list of hits in the PLSDB can be found in file 4 in the supplemental material). One of these previously identified plasmid candidates is plasmid 106_LNODE_1, which had previously been reported by Petersen et al. (28) as pLA6_12. They showed that pLA6_12 is present in many marine samples obtained from locations all around the world, including some samples that were analyzed in this study. The other five plasmid candidates (whose lengths are 1,850 to 2,200 bp) were either fully (>90% coverage) or partially (50 to 70% coverage) aligned to larger plasmids (100 to 750 kbp) (Table 1) in the PLSDB. Two of the candidates (53_LNODE_1 and 281_RNODE_8) contain genes of group II intron mobile elements. Another two candidates (9_LNODE_1 and 276_RNODE_3) contain the IS*1634* family transposase gene.

Searching the candidates against the nonredundant nucleotide database (35) resulted in significant matches (identity percentage $> 65$%) for at least part of the sequence of 201 candidates (55.5%). However, only 9 (2% of the total 362 candidates) were found with coverage and identity percentage above 90% (Table 1). Out of all matches, 99 (49%) were at least partially aligned to bacterial chromosomes, and 82 (40%) were at least partially aligned to viral elements (viruses and phages). Of the 362 candidates, 161 (44.5%) showed no significant match to any entry in the nonredundant nucleotide database, although 140 (87%) of those 161 candidates were found at more than one sampling point in our analysis. Thus, the draft marine plasmidome that we present here is composed mostly of novel sequences.

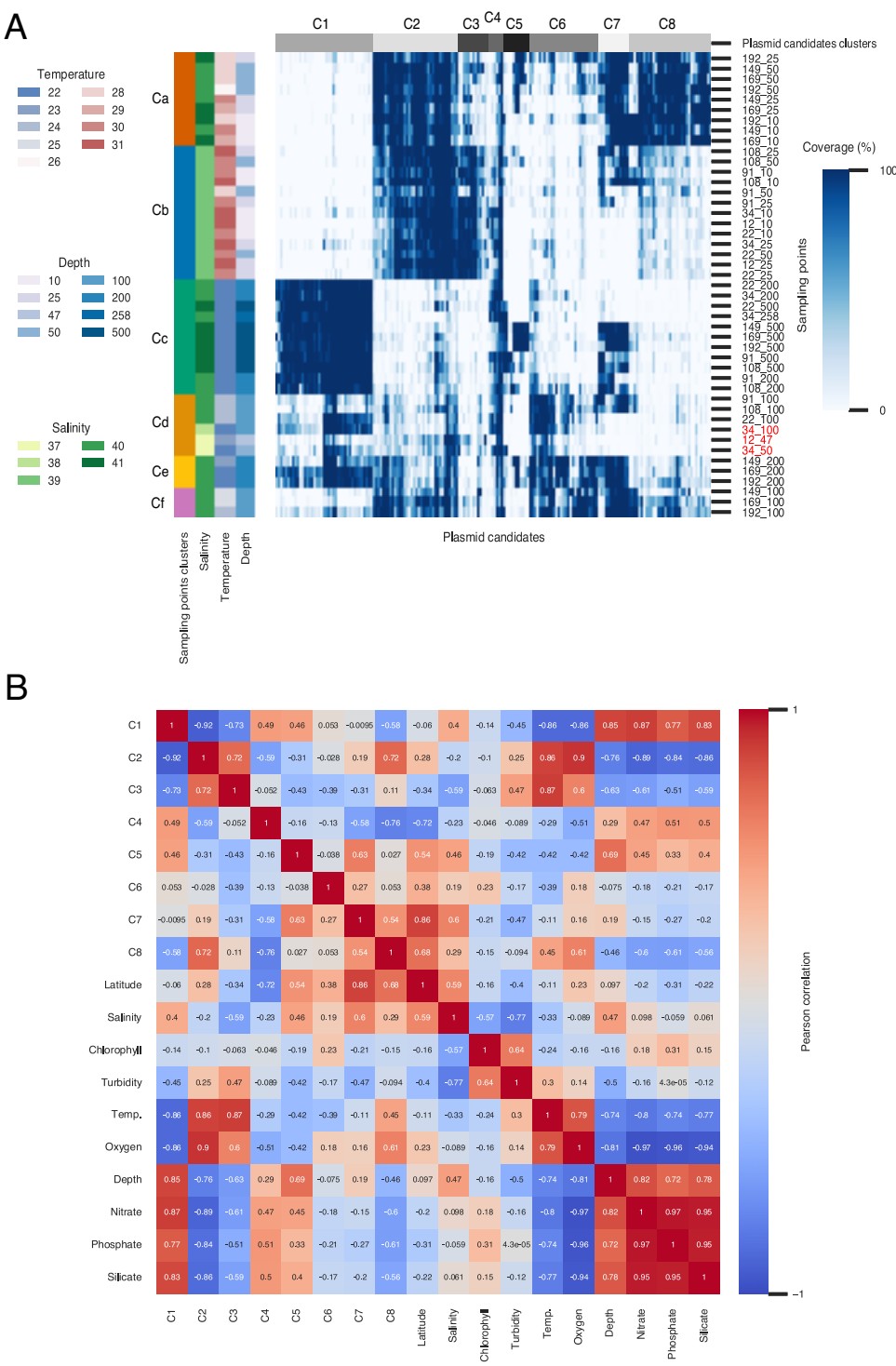

**FIG 2** Distribution patterns of plasmids reflect the physical conditions at the stations. (A) Heat map of the percentage of plasmid length covered in each sampling point. Each column represents a plasmid candidate, and each row represents a sampling point. Sampling point names are in an *x_y* format, where *x* is the station number, as appears in Fig. 1B, and *y* is the depth in meters. Clusters of plasmids are at the top (C1 to C8), and clusters of sampling points are on the left (Ca to Cf). Colored bars on the left show the temperature, depth, and salinity of each sampling point. GAIW sampling points are given in red. Depths (and temperatures) for the clusters are as follows: for the orange cluster (Ca) of northern shallow water sampling points, 50 to 10 m (26.2 to 29.8°C); for the blue cluster (Cb) of southern shallow water sampling points, 50 to 10 m (28.5 to 31.2°C); for the southern relatively deep and cold green cluster (Cc), 500 to 200 m (21.5 to 22.2°C); for the shallower and warmer more southerly ochre cluster (Cd), 100 to 47 m (23 to 24°C), except for station 34, which was colder (21.6°C); for the northern sampling points of the yellow and pink clusters (Ce and Cf), 200 m (22.3 to 23°C) and 100 m (24 to 25°C), respectively (Fig. 1B, clusters Ca to Cf in the 3D representation of the study area). (B) Correlation matrix of plasmid candidate clusters (C1 to C8) and physical parameters. Colors represent positive (dark red) and negative (dark blue) correlations. The numbers inside the cells are Pearson's correlation coefficients.

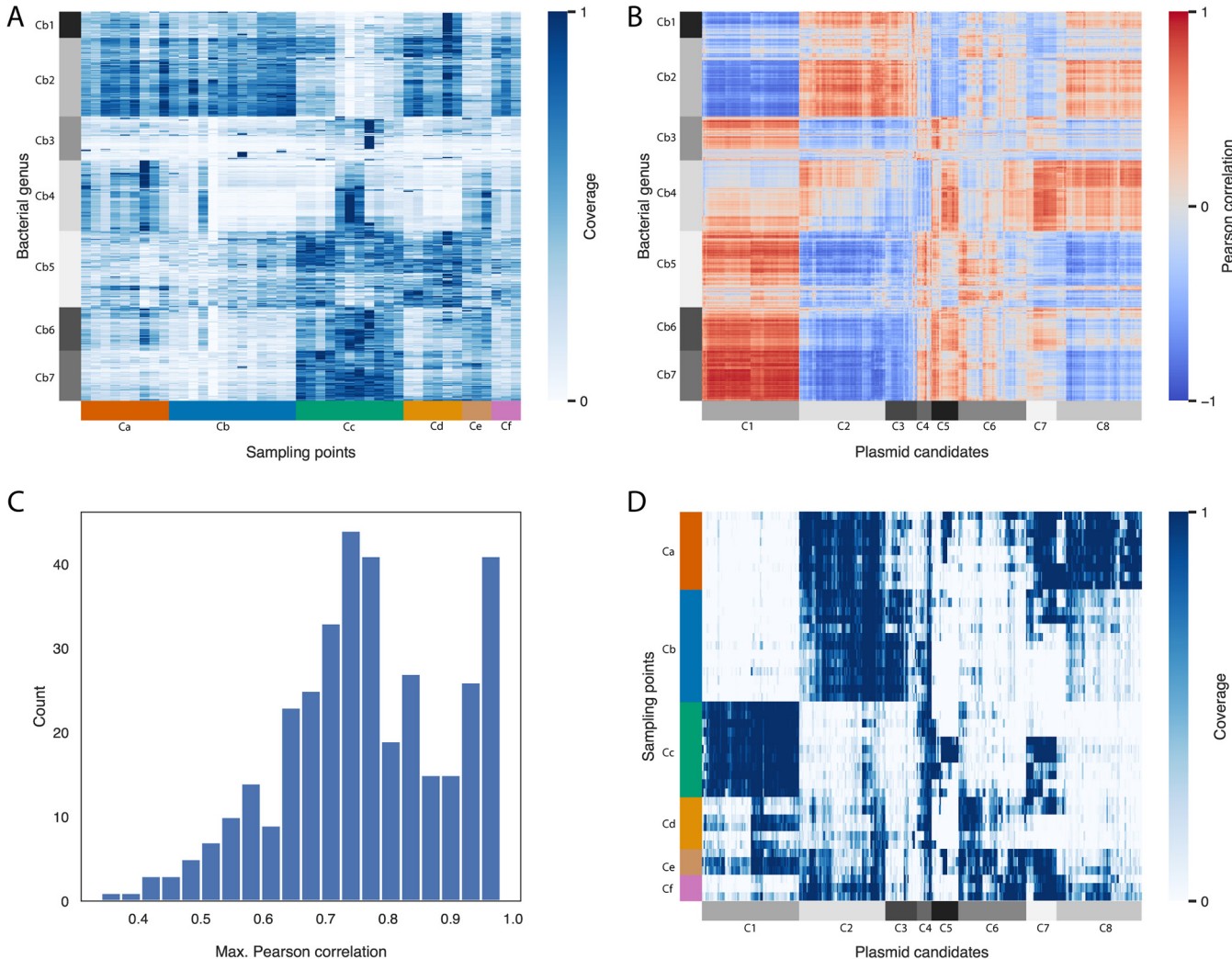

**FIG 3** Association between the distribution patterns of plasmids and microorganisms in the Red Sea. (A) Heat map of the microbial genus presence in each sampling point. Each column represents a sampling point. Sampling points are ordered as in the rows in Fig. 2A and 3D. Each row represents a microbial genus. Clusters of microbial genera are on the left (Cb1 to Cb7), and clusters of sampling points are at the bottom (Ca to Cf). (B) Pearson correlation coefficient matrix between the distribution patterns of microbial genus (rows in panel A) and plasmid candidates (columns in panel D). Colors represent positive (dark red) and negative (dark blue) correlations. Clusters of microbial genera are on the left (Cb1 to Cb7), and clusters of plasmid candidates are at the bottom (C1 to C8). (C) Distribution of maximum Pearson correlation coefficients for each plasmid candidate with microbial genera. (D) Heat map of the percentage of plasmid length covered in each sampling point (similar to Fig. 2A). Clusters of plasmid candidates are at the bottom (C1 to C8).

**Distribution of predicted functions.** As the sequences of the 362 plasmid candidates are mostly novel, we aimed to characterize them computationally. First, we identified between 1 and 752 open reading frames (ORFs) per plasmid, yielding a total of 6,028 ORFs predicted in the entire draft plasmidome (average, 17 ORFs per plasmid; median, 3 ORFs) (Fig. 4A). Thereafter, clusters of orthologous group (COG) categories (36) were assigned to 1,576 (26.2%) ORFs (Table 2) with eggNOG-mapper v.2 (37). However, informative COG categories (other than "unknown") were assigned to only 1,050 (17.4%) ORFs (Table 2; see Fig. S1 and file 5 in the supplemental material). The most highly represented functional categories in the plasmidome were replication (13.1%, COG-L), cell wall/membrane/envelope biogenesis (10.03%, COG-M), and energy production and conversion (8.9%, COG-C). These percentages differ significantly from the frequencies of those categories in the COG database, wherein COG-L represents 2% of the genes (hypergeometric distribution $P$ value = 2.47e−107), COG-M, 2.6% of the genes (hypergeometric distribution $P$ value = 1.23e−46), and COG-C, 4% of the genes (hypergeometric distribution $P$ value = 5.97e−17).

The predicted ORFs were searched in specific databases of resistance gene orthologs, namely, the Antibacterial Biocide and Metal Resistance Genes Database (BacMet) (38) and

**TABLE 1** Plasmid candidates' taxonomy annotation

| Database and plasmid candidate | Alignment length (bp) | % alignment | % identity | Annotation | Overlapping genes | Query length/hit length (bp) |
|---|---|---|---|---|---|---|
| PLSDB[a] | | | | | | |
| 106_LNODE_1 | 4,446 | 78.7 | 99.9 | Roseovarius sp. THAF27 plasmid pTHAF27_d, complete sequence | Replication/maintenance protein RepL, MobA/MobL, MAPEG, TetR/AcrR, type II toxin-antitoxin system PemK/MazF | 5,651/5,571 |
| 53_LNODE_1 | 1,956 | 94 | 76.2 | Vibrio parahaemolyticus strain 2011VPH2 plasmid pVPH2, complete sequence | Group II intron reverse transcriptase/maturase | 2,088/198,487 |
| 281_RNODE_8 | 2,083 | 99.76 | 74.2 | Vibrio alginolyticus strain C1579 plasmid pC1579 | Group II intron reverse transcriptase/maturase | 2,088/236,774 |
| | 1,193 | 57 | 78.6 | Vibrio parahaemolyticus strain 2011VPH2 plasmid pVPH2, complete sequence | Group II intron reverse transcriptase/maturase, hypothetical protein | 2,091/198,487 |
| | 1,196 | 57.2 | 78.5 | Vibrio alginolyticus strain C1579 plasmid pC1579, complete sequence | Group II intron reverse transcriptase/maturase/RNA-directed DNA polymerase, hypothetical protein | 2,091/236,774 |
| 9_LNODE_1 | 992 | 52.2 | 75.5 | Rhodococcus sp. strain DMU1 plasmid unnamed | IS1634 family transposase | 1,900/716,018 |
| 276_RNODE_3 | 1,026 | 55.37 | 73.8 | Rhodococcus jostii RHA1 plasmid pRHL1, complete sequence | IS1634 family transposase | 1,853/1,123,075 |
| 290_RNODE_10 | 1,994 | 91.8 | 74.6 | Acinetobacter baumannii strain VB958 plasmid unnamed1, complete sequence | Alkaline phosphatase D family protein | 2,172/561,419 |
| Nonredundant nucleotide database[b] | | | | | | |
| 4_LNODE_1 | 5,316 | 98 | 99.7 | Prochlorococcus sp. strain RS04\|RS01\|RS50 genome | Glycerol kinase, FAD dependent oxidoreductase, glycosidase | 5,419/1,656,322\|1,657,699\|1,656,133 |
| 53_LNODE_1 | 2,088 | 100 | 99.5 | Alteromonas macleodii strain F12 plasmid unnamed4 | Retron-type reverse transcriptase, hypothetical protein | 2,088/865,153 |
| | 2,089 | 100 | 99.5 | Alteromonas macleodii strain F12 chromosome, complete genome | Retron-type reverse transcriptase, hypothetical protein | 2,088/3,117,849 |
| 299_NODE_4 | 4,063 | 100 | 99.7 | Aeromicrobium erythreum strain AR18, complete genome | Serine protease, WXG100 type VII secretion target, hypothetical proteins | 4,062/3,629,239 |
| 290_RNODE_10 | 2,171 | 99.95 | 98.5 | Acinetobacter venetianus strain TUST-DM21 chromosome, complete genome | Alkaline phosphatase D family protein | 2,172/3,552,418 |
| 269_RNODE_1 | 1,919 | 100 | 100 | Alteromonas macleodii strain D7, complete genome | Alkaline phosphatase | 1,919/4,575,623 |
| | 1,919 | 100 | 99.9 | Alteromonas macleodii strain F12 plasmid unnamed4 | Alkaline phosphatase | 1,919/865,153 |
| 267_RNODE_10 | 1,981 | 100 | 99.95 | Alteromonas sp. strain MB-3u-76 chromosome, complete genome | Cytochrome c, glycyl-tRNA synthetase, $D$-glycero-$\beta$-$D$-manno-heptose 1,7-bisphosphate 7-phosphatase | 1,981/4,360,928 |
| 294_RNODE_9 | 3,400 | 93 | 99.3 | Prochlorococcus sp. strain RS04\|RS01\|RS50 genome | Trehalose-6-phosphate synthase, RNA binding protein, PDZ domain-containing protein, DUF3764 family protein | 3,656/1,656,322\|1,657,699\|1,656,133 |
| 299_NODE_3 | 18,873 | 94.6 | 100 | Aeromicrobium erythreum strain AR18, complete genome | SDR family NAD(P)-dependent oxidoreductase | 19,959/3,629,239 |
| 281_RNODE_8 | 2,095 | 100 | 94.1 | Alteromonas macleodii strain F12 chromosome, complete genome | Group II intron reverse transcriptase/maturase, hypothetical protein | 2,091/3,117,849 |
| | 2,095 | 100 | 93.4 | Alteromonas macleodii strain F12 plasmid unnamed4 | Group II intron reverse transcriptase/maturase, hypothetical protein | 2,091/865,153 |

[a]Above 50% coverage and 70% identity.
[b]Above 90% coverage and identity.

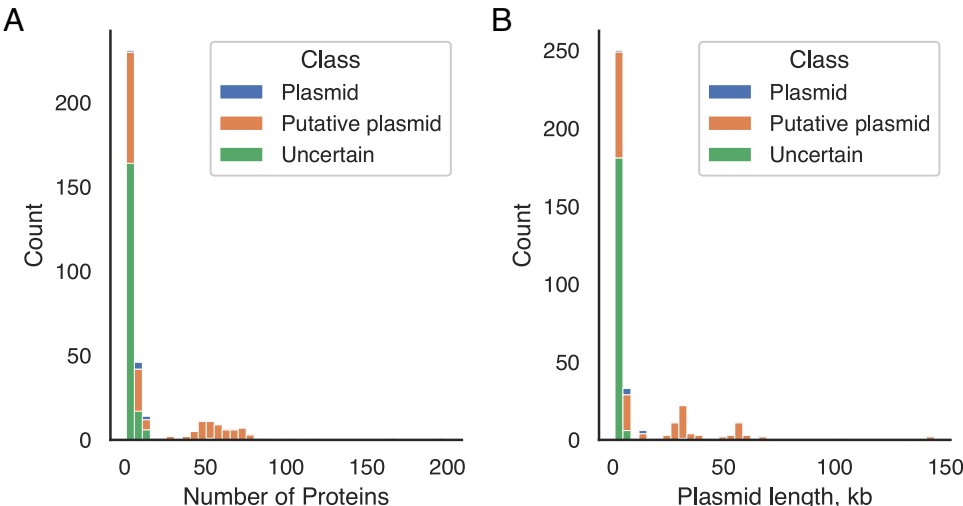

**FIG 4** Distribution of ORFs and plasmidome size distribution. (A) Distribution of number of ORFs per plasmid candidate, truncated at 200. Only one plasmid, 297_RNODE_13, having 752 ORFs, is not shown. Colors represent classification of plasmid candidates. (B) Distribution of the lengths of the plasmid candidates, truncated at 150 kb. Truncation removed plasmid candidate 297_RNODE_13, with a length of 755.6 kb.

the Comprehensive Antibiotic Resistance Database (CARD) (39). Thereafter, to study the association between the distribution of plasmid functional capabilities and the physical properties of the sampling points, we collapsed the functions assigned to the predicted ORFs in the plasmids in each position and considered those functions as representing the functional arsenal at that position. Genes encoding antibiotic resistance were found for 39 sampling points (86.7%) (Fig. 5A). The most frequently carried genes were those encoding resistance to macrolides (86.7% of sampling points), lincosamides (84.4% of sampling points), and streptogramins (84.4% of sampling points). In contrast to the abundance of antibiotic resistance genes, genes encoding metal or biocide resistance were found for only 25 sampling points (55.6%) (Fig. 5B). The most frequently carried metal resistance genes provide resistance to arsenic (42.2% of sampling points), antimony (42.2% of sampling points), and copper (42.2% of sampling points). The overlap between the locations with the highest number of antibiotic resistance genes and the locations with the highest number of metal or biocide

**TABLE 2** Frequency of COG categories

| COG category | Functional category | No. of plasmids | Function frequency | Function frequency without unknown function |
|---|---|---|---|---|
| S | Function unknown | 526 | 31.63 | |
| L | Replication, recombination and repair | 149 | 8.96 | 13.1 |
| M | Cell wall/membrane/envelope biogenesis | 114 | 6.86 | 10.03 |
| C | Energy production and conversion | 101 | 6.07 | 8.88 |
| E | Amino acid transport and metabolism | 92 | 5.53 | 8.09 |
| K | Transcription | 87 | 5.23 | 7.65 |
| P | Inorganic ion transport and metabolism | 75 | 4.51 | 6.6 |
| J | Translation, ribosomal structure and biogenesis | 71 | 4.27 | 6.24 |
| I | Lipid metabolism and transport | 63 | 3.79 | 5.54 |
| O | Posttranslational modification, protein turnover, chaperones | 62 | 3.73 | 5.45 |
| G | Carbohydrate metabolism and transport | 60 | 3.61 | 5.28 |
| Q | Secondary metabolite biosynthesis, transport, and catabolism | 55 | 3.31 | 4.84 |
| F | Nucleotide metabolism and transport | 46 | 2.77 | 4.05 |
| H | Coenzyme metabolism and transport | 44 | 2.65 | 3.87 |
| T | Signal transduction | 38 | 2.29 | 3.34 |
| U | Intracellular trafficking, secretion and vesicular transport | 36 | 2.16 | 3.17 |
| D | Cell cycle control, cell division, chromosome partitioning | 18 | 1.08 | 1.58 |
| V | Defense mechanisms | 18 | 1.08 | 1.58 |
| N | Cell motility | 6 | 0.36 | 0.53 |
| W | Extracellular structures | 1 | 0.06 | 0.09 |
| Z | Cytoskeleton | 1 | 0.06 | 0.09 |

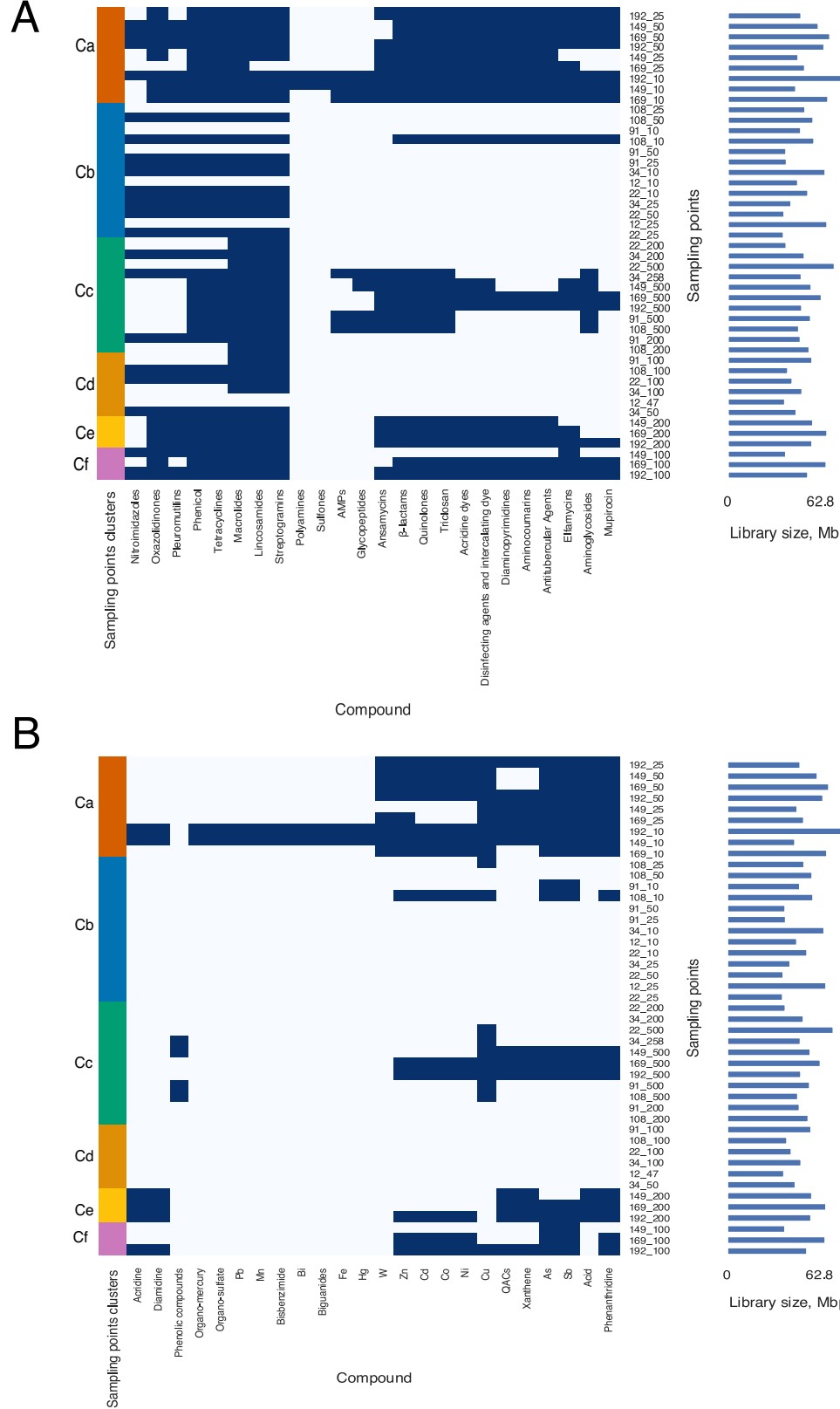

**FIG 5** Distribution of resistance to compounds at the sampling points. Presence of resistance to (A) antibiotics (according to CARD [39]) and (B) metals and biocides (according to BacMet [38]) is represented by heat maps. Each row represents a sampling point (x_y, where x is the station number and y is the station depth in meters), and each column represents resistance to a specific compound. Presence of a resistance encoding gene is shown in dark blue; absence, in white. Bar plots on the right illustrate library sizes at each sampling point. Sampling points are ordered as in Fig. 2A.

**TABLE 3** Plasmid candidates classified as "plasmids"

| Plasmid | Size (bp) | % GC | MOB group | Predicted genes and proteins with unknown functions |
|---|---|---|---|---|
| 1_LNODE_1 | 13,274 | 59.98 | MOBF | Type IV secretion system, relaxase domain-containing protein, recombinase family protein, HTH, restriction endonuclease, *S*-adenosylmethionine-binding protein, AAA family ATPase, *repA* genes; 3 hypothetical proteins |
| 3_LNODE_1 | 5,082 | 37.74 | MOBQ | MobA/MobL, mobilization protein, RepM/RepB genes; 3 hypothetical proteins |
| 30_LNODE_1 | 11,671 | 29.28 | | Integrase, restriction endonuclease, 2 restriction endonucleases, resolvase, phospholipase D genes; 2 hypothetical proteins |
| 39_LNODE_1 | 3,597 | 53.85 | | MobC, HTH, repA genes; 2 hypothetical proteins |
| 106_LNODE_1 | 5,651 | 59.3 | MOBQ | *repL*, MobA/MobL, MAPEG, TetR/AcrR, type II toxin-antitoxin system PemK/MazF; 3 hypothetical proteins |
| 297_RNODE_15 | 6,231 | 62.94 | | Transferase activity, antitoxin, GNAT family *N*-acetyltransferase, ParB/RepB genes; 2 hypothetical proteins |
| 298_RNODE_5 | 6,614 | 54.45 | | Site-specific DNA methyltransferase, recombinase, Tn*3* family transposase, VapB and VapC genes; 3 hypothetical proteins |

resistance genes is worthy of attention. These locations—constituting cluster Ca (in particular, 192_10 and 149_10) (Fig. 5A and B)—are at the northerly positions of the cruise track (Fig. 1B). It is unlikely that the multiple functions at these locations can be explained solely by larger library sizes (right bars in Fig. 5A and B). A more probable explanation is that the geographical location and the environmental conditions at each sampling point define a minimal set of functions to support the local bacterial community, with these functions being at least partially carried by the plasmids.

Overall, 3,062 of 6,028 ORFs (50.8%) were assigned to known functions with eggNOG-mapper (37) and InterProScan (40) in the BacMet database (38), CARD (39), and A Classification of Mobile Genetic Elements (ACLAME) database (41) (see file 6 in the supplemental material).

**Plasmidome classification.** As the computational tools used in this study may still report nonplasmidic cyclocontigs as plasmids (30), we applied additional classification to assess the likelihood of a plasmid candidate being truly plasmidic. For this classification, we used the virus verification tool viralVerify (42), since this tool was shown to be better at differentiating between plasmids and viruses than the plasmid verification tool plasmidVerify (30). viralVerify (42) classified six candidates as "plasmids," based on their functional repertoire. For further characterization and classification of the candidates, we then searched for functional assignment of plasmid-associated genes. We found orthologs to the predicted ORFs in the ACLAME database (41), with eggNOG-mapper v.2 (37) and InterProScan (40). Based on the list of plasmid-associated functions, candidate 3_LNODE_1, harboring replication initiation and MOBA/MOBL family proteins, was added to the candidates classified as "plasmids," and 167 plasmid candidates were classified as "putative plasmids." The remaining 188 candidates, which do not harbor any plasmid-associated genes or for which all genes have unknown functions, were classified as "uncertain" (see files 2, 5, and 6 in the supplemental material). Plasmid candidates classified as "uncertain" or "plasmid" tend to bear up to 20 ORFs (Fig. 4A). In contrast, plasmid candidates classified as "putative plasmid" have various numbers of ORFs, with the number correlating with the candidate's size.

Table 3 lists the seven candidates classified as "plasmids" and their predicted genes. Each candidate that was classified as a "plasmid" harbors plasmid-associated genes, e.g., replication control genes (such as RepA/L/M), conjugation or mobility genes (MobA/L/C, relaxase), partition genes (ParB/C), toxin-antitoxin genes, or resistance genes. Each of these plasmids also has proteins with unknown functions. Only three of these plasmids were assigned to a MOB group (1_LNODE_1 to MOBF and 3_LNODE_1 and 106_LNODE_1 to MOBQ), and none was assigned to a plasmid taxonomic unit (PTU) by COPLA (43).

The lengths of most plasmid candidates (358/362 [98%]) did not exceed 100 kb (Fig. 4B). The lengths of all the candidates classified as "plasmids" fell in the range of 4 to 20 kb. All candidates classified as "uncertain" were shorter than 32 kb, and most of them (171/188 [90%]) were even shorter than 4 kb. Plasmid candidates classified as "putative plasmids" varied in length. The large candidates contained a large number of ORFs, some of which

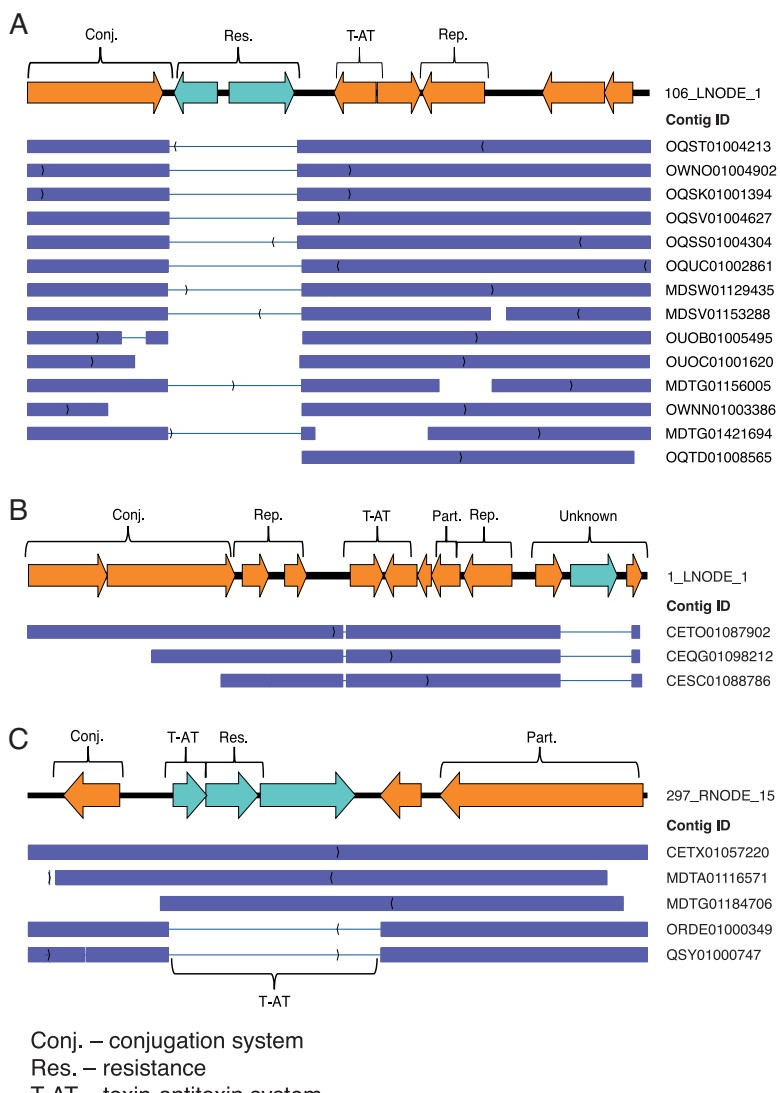

Conj. – conjugation system
Res. – resistance
T-AT – toxin-antitoxin system
Rep. – replication system
Part. – partition system

**FIG 6** Global plasmid distribution. Three plasmids identified in our study, 106_LNODE_1 (A), 1_LNODE_1 (B), and 297_RNODE_15 (C), are shown. In each panel, plasmids are shown at the top; orange arrows represent functional essential genes, and turquoise arrows represent interchangeable cassettes. Alignment of contigs with similar sequences in other marine metagenome studies is shown at the bottom, and the GenBank accession numbers of those contigs are presented on the right.

might be plasmid-related genes and others of which are typical virus or chromosomal genes; therefore, it was not surprising that they fell into the "putative plasmid" category.

**Global distribution of plasmids.** Three of the seven candidates classified as plasmids in this study were found in other locations around the world (see file 7 in the supplemental material). Plasmid 106_LNODE_1, a version of pLA6_12, which was previously reported by Petersen et al. (28), was previously found in the Indian, Pacific, and Atlantic oceans (coverage > 50%, identity > 90%) (Fig. 6A). Plasmid 1_LNODE_1 was similar (coverage > 50%, identity > 80%) to assembled contigs CETO01087902 (found in the South Pacific Ocean), and CEQG01098212 and CESC01088786 (found in the South Atlantic Ocean) (Fig. 6B). All these contigs contain the same backbone, with high similarity in the replication, conjugation, toxin-antitoxin, and partition regions, but they have different cassettes of hypothetical proteins. The two assemblies from the Atlantic Ocean lack the conjugation-associated gene (type IV secretion system gene) in the mobility gene region. Contigs similar (coverage > 50%, identity > 80%) to candidate 297_RNODE_15 were detected in

the Indian (MDTA01116571), Pacific (CETX01057220 and MDTG01184706), and Atlantic (ORDE01000349 and QSY01000747) oceans (Fig. 6C). In this case, too, the backbone genes are highly similar, but the two contigs from the Atlantic Ocean contain a cassette that differs from 297_RNODE_15 for the toxin-antitoxin system, with YhaV as the toxin and PrlF as the corresponding antitoxin (T-AT region in Fig. 6C). Finally, the ORF associated with the conjugation system is missing in contig MDTG01184706 from the Pacific Ocean.

## DISCUSSION

In this study, we reanalyzed metagenomic sequences from the Red Sea (32), with the aim of detecting and characterizing plasmids of marine origin. Although the isolated DNA had not been enriched for plasmids, we detected 362 plasmid candidates from 45 sampling points. We note that it is estimated that in most cases, plasmids can be detected only if plasmid extraction procedures are performed (5, 13). Moreover, the data set used in this study was collected from the 0.1- to 1.2-$\mu$m fraction, which excludes most of the eukaryotes, such as diatoms, which were shown to contain plasmids (44). Thus, these candidates constitute just the tip of the iceberg of the marine plasmidome in that region.

The physical properties (depth and temperature) of the sampling points were shown to correlate with the distribution patterns of the plasmid candidates. While other studies have reported temperature as a key predictor of microbial diversity in the oceans (45–47), the physical properties of the sampling points are better reflected by the distribution of plasmids than by the microbial distribution at the genus level (Fig. 3A) (34). Interestingly, in terms of plasmid composition, all three GAIW sampling points were coclustered with three sampling points that are not GAIW (Fig. 2A, cluster Cd). The GAIW sampling points, where the temperatures are lower (21 to 23°C) than those of the Red Sea, were clustered with warmer sampling points (~24°C). Thus, at least in this case, the effect on the plasmidome of the incoming stream is stronger than that of the water temperature.

The distribution patterns of some plasmid candidates strongly correlate with the distribution patterns of at least one microbial genus (Fig. 3B). Such strong correlation between the distribution patterns indicates colocalization and may suggest a potential host for the plasmid candidate. On the other hand, the distribution patterns of other candidates are not strongly correlated with the distribution patterns of any genus (Fig. 3B), which may suggest that these candidates are broad-host-range plasmids. However, the potential assignment of host and host range based on colocalization is hypothetical and should be further tested and validated. Notably, one plasmid candidate cluster (C1) strongly correlates (Pearson correlation coefficient of 0.97) with one microbial cluster (Cb7). Both the microbial and the plasmid candidate cluster distribution patterns are higher in deep-water environments (Fig. 3A and D, respectively) and may correspond to unique populations of microbes and plasmids which are specific to deep waters. Thus, we hypothesize that C1 plasmids provide deep water adaptation, and it is possible that their distribution pattern is attributed to a single deep-water genus. In contrast, the distribution pattern of C6 plasmids, for example, does not strongly correlate with a single microbial cluster (Fig. 3B) and thus cannot be attributed to a single genus, and those plasmids are hypothesized to have a broad host range.

Most plasmid candidates (90.61%) were present at more than one sampling point, suggesting that they represent a reliable sequence and not an assembly artifact. Partial coverage of some plasmids at some sampling points was also revealed, indicating either conserved regions that are shared between different plasmids—be it a shared backbone (namely, the same plasmid with a different cassette) or a shared functional gene—or plasmids at low copy numbers that are not fully covered within the sequencing data. This pattern of conservation of plasmid sequences across the samples was previously identified by Kothari et al. (13), who attributed ecological significance to plasmids in terms of maintenance and transfer of conserved key functionalities in an ecosystem, based on reports of the plasmidome in soil (48) and rumen (5) environments. We demonstrated this pattern locally with our draft plasmidome and globally in the case of three

plasmids, 106_LNODE_1, 1_LNODE_1, and 297_RNODE_15, only one of which has been previously reported (28).

Only 3,062 of the predicted ORFs (50.8%) matched proteins with known functions. Previous plasmidome annotation studies identified 25 to 61% of ORFs in lakes (6), a wastewater treatment plant (49), and rumen (5, 18), and our results are thus in agreement with previously reported studies. Nonetheless, despite the low number of annotated plasmids from marine sources in the plasmid database and the reported differences between marine and terrestrial plasmids, the ability to match 50.8% of the predicted ORFs with a known protein is encouraging and serves as a good starting point for the identification and annotation of marine plasmids.

The genes encoding DNA replication, recombination, and repair (COG category L) were the most represented genes in the marine plasmidome—significantly more than in the COG database. Since replication is required for plasmid survival and is one of the main backbone plasmidic functions, genes involved in replication and transposable elements are highly abundant in the plasmids (50). The other functional categories that are highly represented in this study (cell wall/membrane/envelope biogenesis, energy consumption and conversion, posttranslational modifications and protein turnover, carbohydrate metabolism and transport, amino acid transport and metabolism, inorganic ion transport and metabolism, intracellular trafficking, secretion, and vesicular transport) are also similar to those in previous plasmidome studies (5, 6, 13, 49). These functional categories mostly represent functions that are part of the plasmid maintenance and survival functional repertoire (replication, conjugation, and resistance) and are thus expected to be represented in most plasmids (13).

The assignment of functions to the ORFs allowed us to identify seven candidates as most probably plasmids and 167 candidates as putative plasmids, based on the genes that they carry. Unlike bacterial chromosomes, plasmids are very dynamic and do not have specific markers, as some plasmids are known to integrate into the host genome and transfer sections of the chromosome along with their conjugative machinery into a recipient cell (51). Such integrations occur via homologous or nonhomologous recombination, allowing fast and large evolutionary jumps within the affected genes (52). Thus, the fact that a candidate plasmid contains chromosomal or viral genes, or a partial match to chromosomal or viral DNA (as was observed for 158 of 362 candidates [43%]), does not rule out the possibility that these candidates are plasmids, as it is common for plasmids to exchange DNA with bacterial chromosomes and viruses (52).

Generally, plasmids vary markedly in size, with the smallest being around 846 bp (and carrying only the replication initiation gene) and some plasmids exceeding the size of some bacterial chromosomes (and carrying ≥1,000 genes) (53). Associations between plasmid and bacterial genome size and between plasmid size and the mobility genes it carries (if any) were shown by Smillie et al. (53). Sizes of candidates classified in this study as definitely "plasmids" were in range of 4 to 20 kb, which is a typical size for both mobilizable and nontransmissible plasmids (43). The distribution of candidate sizes showed peaks at around 5, 30, and 60 kb, which is similar to the distribution of plasmids described by Smillie et al. (53). It is important to mention that candidates' size range in this study might not represent the real plasmid size range, due to the limitations of the sequencing and bioinformatic and microbial extraction methods (13, 30).

As mentioned above, marine plasmids differ from terrestrial plasmids in terms of their DNA sequence and carried genes and are underrepresented in the PLSDB. In accordance, matches to the PLSDB (22) were found for only a few plasmid candidates (1.65%). One of the hits was to a plasmid previously isolated from *Roseovarius* sp. strain THAF27 from a marine aquarium sample (GenBank accession no. CP045397.1). That plasmid was characterized as a cassette-containing pLA6_12-like plasmid, restricted to marine environments (28). Interestingly, the other hits were for plasmid candidates with relatively short sequences, which aligned to larger plasmids (Table 1). The annotation of the ORFs in these overlapping regions indicates that most of these short candidates were transposons, but it was not clear whether they exist as stand-alone mobile genetic elements or whether they are part of other

plasmids or chromosomes and were found by our pipeline by virtue of repeats at the ends of the transposon. Some of these short candidates were classified by us as "uncertain" due to the lack of plasmid-associated genes. Nevertheless, plasmid candidates that were matched to plasmids in the PLSDB were previously isolated from strains of *Roseovarius* species, *Vibrio parahaemolyticus*, *Vibrio alginolyticus*, *Rhodococcus* species, and *Acinetobacter baumannii*. *Vibrio* and *Roseovarius* are typical of marine environments, whereas *Rhodococcus* is mostly known as a soil bacterium. However, some *Rhodococcus* species were previously found in marine environments (54). Interestingly, although *Acinetobacter baumannii* is a soil bacterium, it was shown that the Rep proteins of many *Synechococcus* plasmids are similar to those of plasmids from several strains of *Acinetobacter baumannii* (20). Hence, it is not surprising to find matches to plasmids from *A. baumannii* in our study as well.

In addition to 106_LNODE_1, which is the local version of the previously reported hitchhiker plasmid pLA6_12 (28), we found "relatives" with high sequence similarity to two more plasmid candidates that were classified as "plasmid" by us in other whole-genome sequences from available marine metagenomes. Plasmid 1_LNODE_1, which in our analysis was *de novo* assembled and revealed by mapping for 19 different sampling points, and plasmid 297_RNODE_15, which was found at three sampling points in our analysis, were also partially detected in biosamples collected in the Tara Oceans expedition 2009 to 2013 and the Malaspina Expedition 2010 in different locations in the Atlantic, Pacific, and Indian oceans. This global distribution with different functional cassettes suggests that these plasmids are potential vectors able to transport different integration cassette motifs across vast geographic distances. The dynamics of interchangeable cassettes provides evidence for horizontal gene transfer events, which support the classification of these candidates as plasmids. The observed variable regions in the plasmids contain genes encoding either resistance to toxins or various toxin-antitoxin systems or lack mobility genes, which might shed light on rapid adaptation to environmental pollutants.

The plasmid candidates found in 45 and 31 of the 45 stations carried antibiotic resistance and metal resistance genes, respectively, in accordance with antibiotic resistance and metal resistance being functional capabilities frequently supplied by the plasmidome (13, 55). Even though previous studies found no evidence for resistance to metals or biocides being associated with contamination (28), we showed that both metal and antibiotic resistance genes were enriched for specific sampling points, suggesting that there is some association between the environmental conditions and the development of resistance. For example, plasmid 106_LNODE_1 in our study (a version of pLA12_6) includes the MAPEG cassette, providing protection against xenobiotics and/or oxidative stress (56), which have been reported to be typical environmental stressors in the Red Sea (28). This plasmid is found in many other locations around the globe, but with a different cassette of functional genes at each location, suggesting adaptation of the variable cassette to the specific conditions of the location in which the plasmid resides. Thus, our analysis of the global distribution of plasmids found in this study shows that plasmids provide site-dependent phenotypic modules to their ecological niches, in accordance with previous studies (28, 57).

In this study, we reanalyzed publicly available metagenomic sequencing data from the Red Sea and demonstrated that marine plasmids can be discovered and characterized from publicly available metagenomics data that have not been enriched for plasmids. In addition, we showed site-dependent plasmid distribution and correlations between plasmid distribution patterns and environmental conditions, demonstrating the importance of plasmids in the microbial ecosystem. We detected seven definite plasmids and 355 other candidate plasmids for which additional research is required to validate their plasmidic nature. To increase the representation of environmental marine plasmids in plasmid databases, additional studies on plasmid detection in marine environments should be performed. Our study gives only a glimpse into the marine plasmidome, probably one of the largest and most untapped sources of genes with novel functions.

## MATERIALS AND METHODS

**Data and assembly.** Raw Illumina sequencing paired-end reads were obtained from the NCBI BioProject database PRJNA289734 (32)—overall, 45 sampling points from different depths at eight stations

(Fig. 1B), for which the small microbial fractions (between 0.1 to 1.2 $\mu$m) were sequenced. Metagenomic reads from each sampling point were assembled using SPAdes 3.14 (33). Plasmids were identified by Recycler (17) and metaplasmidSPAdes (30), both with default parameters. Plasmid candidates that were identified at the same sampling point by both methods were detected by pairwise alignment of predicted plasmids from both Recycler and metaplasmidSPAdes using BLASTN (58) with a $10^{-3}$ E-value cutoff. Duplicates (candidates with at least 99% identity and 99% coverage) were removed, yielding a total of 736 plasmid candidates for all sampling points together (Fig. 1A; see file 8 in the supplemental material).

**Plasmidome definition.** All plasmid candidates identified at all sampling points were aligned to each other using BLASTN (58). By using Python (3.10) scripts, the alignment percentage (AP) between each pair of candidate plasmids was calculated as the ratio between the length of the alignment and the length of the shorter member of the pair. In cases in which more than one significant alignment was identified along the two plasmids, the total length of those alignments (excluding overlaps) was considered the alignment length. To detect highly similar plasmid candidates, the AP matrix was used as the input (instead of the correlation matrix) for hierarchical clustering (scipy.cluster.hierarchy.linkage, scipy.cluster.hierarchy.fcluster) (Fig. S2). Following clustering, plasmids with 99% coverage and identity were combined into groups. One plasmid from each group was selected as a representative of that group in the collection of unique plasmids, termed the plasmidome. Plasmid candidates shorter than 1,000 bp were not included in the plasmidome, since short elements are less likely to carry genes, but they were included in the full list of candidates (see file 8 in the supplemental material).

**ORF prediction.** Prodigal (59) was used to detect ORFs in the plasmids. As circular plasmid sequences appear linear in FASTA format, two consecutive plasmid sequences were used as input, to enable identification of ORFs that fall on the arbitrary start/end position of the plasmids. In plasmids that have an ORF spanning their entire length, the ORF appears to exceed the actual plasmid length, and thus, ORFs were truncated to the plasmid length. In ORFs that span the arbitrary plasmid end, the ORF end position is higher than the plasmid end. Thus, the plasmid length is subtracted from the ORF end position. Smaller ORFs which are located on the same strand, start at the 1- to 3-bp position of the plasmid's arbitrary start, lack the start codon, and overlap ORFs that span the arbitrary plasmid end and end at the same coordinate were removed, since partial ORFs were contained in the longer ORFs. To remove any remaining duplicate ORFs, the ORFs in each plasmid were grouped by strand direction (1 or $-1$). For each strand, the overlapping ORFs were extracted, and from these overlapping sequences the smaller ones were removed.

**Mapping the distribution of plasmid candidates.** As the methods for *de novo* assembly of a plasmid require a relatively high coverage of the plasmid in a sample, a plasmid with a relatively low coverage might be missed by our plasmid identification protocol. However, such a plasmid with a relatively low coverage can be detected in a sample when one specifically looks for its presence. Thus, to test for the presence of each plasmid candidate at each sampling point, the raw reads from each sampling point were mapped to the plasmidome using Bowtie2 v. 2.2.5 (60, 61) with default parameters. The plasmid presence matrix records the coverage of each plasmid candidate at each sampling point. In the plasmid presence matrix, both rows (sampling points) and columns (plasmid candidates) were clustered by hierarchical clustering (seaborn.clustermap, scipy.cluster.hierarchy.linkage). A candidate was considered present at a sampling point if the coverage of the candidate at that sampling point was higher than 99%, i.e., more than 99% of the sequence of the candidate was covered by reads.

To calculate the correlation of candidates' clusters with physical parameters at the sampling points, the average coverage of each cluster of plasmid candidates at each sampling point was calculated. Then, a Pearson correlation (stats.pearsonr) was calculated for each cluster's coverage vector and each physical parameter's vector.

**Microbial distribution association with plasmids.** The relative abundance table of microbial data at the genus level was obtained from the study conducted by Thompson et al. (34) and had been scaled using preprocessing. MinMaxScaler to minimize potential bias caused by differences in the original abundance values.

The microbial distribution patterns were clustered hierarchically by seaborn.clustermap and scipy.-cluster.hierarchy.linkage. Pearson correlation coefficients between the presence patterns of plasmid candidates and microbial genera were calculated.

**Plasmidome annotation.** To test whether the plasmids were already known, each plasmid was searched in the NCBI nucleotide database (35) and the PLSDB (22) using BLASTN with a $10^{-3}$ E-value cutoff and 50% coverage. We used COPLA (43) to predict PTUs and MOB groups where possible.

To identify proteins that are typical of plasmids, the COG category for each predicted ORF (36) was predicted using EggNOG (37) with default parameters and InterProScan (40). To test whether COG categories were enriched in our plasmidome compared to the COG database, excluding genes with unknown functions (COG-S), a hypergeometric distribution was used (scipy.stats.hypergeom.cdf and scipy.stats.hypergeom.sf), where $N$ is the number of genes in the COG database, excluding COG-S (9390 genes); $K$ is the number of genes assigned a particular COG category; $n$ is the number of plasmidome ORFs assigned any COG category, except for COG-S (1050 ORFs); and $k$ is the number of plasmidome ORFs assigned a particular COG category. The peptides encoded by the predicted ORFs were searched for known sequence motifs in the BacMet database (38), the Toxin Antitoxin Database (TADB) (62), the ACLAME database (41) and the CARD (39) using BLASTP with a $10^{-3}$ E-value cutoff. To identify resistance to a metal or a biocide compound, the accession number of each hit in BLASTP output was mapped to the BacMet database (38) description. The drug class of antibiotic resistance genes was assigned on the basis of the CARD (39) description. A gene presence matrix, representing the functional capabilities of the plasmidome at each sampling point, was generated. In this matrix, gene presence in any of the plasmids present at the station was defined as 1, while absence in all plasmids present at the station was defined as 0.

**Assignment of confidence level to plasmid candidates.** Plasmid candidates were classified on the basis of the predictions of ORF functions (file 5 in the supplemental material), plasmidVerify (30), and viralVerify (42). Candidates were classified into three categories. (i) The first was plasmids, i.e., candidates predicted as "plasmid" by viralVerify (42). (ii) The second was putative plasmids, i.e., candidates that were not classified as plasmids by viralVerify (42) but met one of the following criteria: (a) any of the candidate's ORFs functions predictions included the word "plasmid"; (b) plasmidVerify (30) predicted the candidate as "plasmid"; (c) any of the ORFs of the candidate was assigned an ACLAME (41) function; or (d) any of its ORF function predictions was a plasmid-associated function, namely, a function that is found on plasmids as defined above in item 1 and putative plasmids defined in item 2a. The third category was "uncertain," i.e., the remaining candidates.

After initial classification, plasmids and putative plasmids were manually reviewed and, when appropriate, reclassified (e.g., candidates harboring more than one plasmid-associated function predicted by at least two function prediction tools were classified as "plasmid").

**Plasmid global distribution.** To reveal the global distribution of candidates classified as "plasmids," each "plasmid" was searched with the BLASTn (35) web version in the public whole-genome shotgun contigs database limited to the marine metagenome (taxid 408172), as was done by Petersen et al. (28). Hits with >50% coverage and >70% identity were considered similar. For each hit, regions that were not matched were searched with BLASTx (35) to provide an annotation for the coding sequences in those regions.

The complete pipeline with Python scripts can be found at https://github.com/Tal-Lab/Plasmidome.

## SUPPLEMENTAL MATERIAL

Supplemental material is available online only.
**SUPPLEMENTAL FILE 1**, XLSX file, 2.2 MB.
**SUPPLEMENTAL FILE 2**, XLSX file, 0.03 MB.
**SUPPLEMENTAL FILE 3**, XLSX file, 3.2 MB.
**SUPPLEMENTAL FILE 4**, XLSX file, 1.1 MB.
**SUPPLEMENTAL FILE 5**, XLSX file, 0.3 MB.
**SUPPLEMENTAL FILE 6**, XLSX file, 0.5 MB.
**SUPPLEMENTAL FILE 7**, XLSX file, 0.01 MB.
**SUPPLEMENTAL FILE 8**, XLSX file, 3.5 MB.
**SUPPLEMENTAL FILE 9**, PDF file, 0.2 MB.

## ACKNOWLEDGMENTS

This study was supported (in part) by grant no. 3-17700 from the Office of the Chief Scientist, Israel Ministry of Health. L.A. is the recipient of a Hi-Tech, Bio-Tech, and Chemo-tech fellowship of Ben-Gurion University of the Negev.

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
