## [Reviewer comments · Microbiology Spectrum]

Microbiology Spectrum

Characterization of the Environmental Plasmidome of the Red Sea

Lucy Androsiuk, Tal Shay, and Shay Tal

Corresponding Author(s): Shay Tal, Israel Oceanographic and Limnological Research Institute

Review Timeline:

Submission Date:	January 26, 2023
Editorial Decision:	March 27, 2023
Revision Received:	May 31, 2023
Accepted:	June 13, 2023

Editor: Tino Polen

Reviewer(s): Disclosure of reviewer identity is with reference to reviewer comments included in decision letter(s). The following individuals involved in review of your submission have agreed to reveal their identity: Daniel Kurth (Reviewer #1)

Transaction Report:

DOI: <https://doi.org/10.1128/spectrum.00400-23>

March 27, 2023

Dr. Shay Tal
Israel Oceanographic and Limnological Research Institute
National Center for Mariculture
P.O.Box 1212
Eilat 88112
Israel

Re: Spectrum00400-23 (Characterization of the Environmental Plasmidome of the Red Sea)

Dear Dr. Shay Tal,

thank you for submitting your manuscript to Microbiology Spectrum.

I have received comments on your manuscript from two experts who suggested modifications. I do hope you will find the reviewers' comments below helpful and look forward to receiving a revised version from you.

Link Not Available

Sincerely,

Tino Polen

Journals Department
Reviewer comments:

Reviewer #1 (Comments for the Author):

This study presents the identification of plasmids based on metagenomic analysis of published data from the Red Sea. The work is very solid and I only found a small issue that was not adequately explained:
When you "merged" the output from Recycler and metaplasmidSPAdes, you found duplicates by local alignment with BLASTN. I doubt you had exactly the same result from both methods. If you didn't, what would you call "duplicates"? Sequences with >90% ID? Please clarify.

Another minor issue is your GitHub address. <https://github.com/Tal-Lab/Plasmidome> is not public yet?

Other than that I enjoyed reading your work. As a minor suggestion, in Fig 3 and 5, I found relatively useless panel A with the full distributions, as plasmids with many ORFs or longer than 150kb are barely seen. I'd rather keep panel B and mention in the legend that XX counts were beyond the range shown.

Reviewer #2 (Comments for the Author):

In this paper the authors analyze publicly available marine metagenome data to search for plasmids. The work develops a pipeline using two previously developed tools and a consensus approach. (Figure 1). The results are interesting but some of the putative plasmids could be chromosomal fragments etc as discussed in Antipov et al (ref 26).

Other comments:

1. I would have appreciated it if the authors did more ground-truthing of their approach, for example bioinformatically spiking in known plasmid reads (fragments) to test assembly and sensitivity to plasmid abundance.
2. The authors need to provide more information and preferably statistical analysis of the relationship of plasmids to bacteria actually present (as seen in metagenome assemblies). The authors could compare the plasmids frequencies/presence to the presence of specific taxa that can be obtained from ref 28 or related papers.
3. The authors need to provide more information about the samples. Specifically, the filter sizes as this is highly relevant to the interpretation of the results." Raw Illumina sequencing paired-end reads were obtained from the NCBI BioProject database PRJNA289734 (ref 28)"
4. The authors should provide %GC about the putative plasmids.
5. Line 327 "Nevertheless, plasmid candidates that were matched to plasmids in the PLSDB were previously isolated from strains of *Vibrio parahaemolyticus*, *V. alginolyticus*, *Rhodococcus* species, and *Acinetobacter baumannii*. All these bacteria are representatives of the common marine environment phyla Proteobacteria and Actinobacteria (50)."

While the *Vibrios* are marine bacteria, *Acinetobacter* and *Rhodococcus* are soil bacteria. This statement is misleading in that the categories of Proteobacteria and Actinobacteria are too broad to be relevant. As seen in ref (19) it states that "Most replication proteins hit by the reads of SynMeta03, SynMeta04, SynMeta05 and SynMeta06 were similar to those of plasmids from several strains of *Acinetobacter baumannii*". There is a distinct possibility that the detected plasmids are from *Synechococcus* or that there is something highly promiscuous about these plasmids. It would be best to use this information to better phrase the results.

References

6. A reference to marine eukaryotic plasmids is
Plasmids in diatom species.

M Hildebrand, D K Corey, J R Ludwig, A Kukel, T Y Feng, and B E Volcani
J Bacteriol. 1991 Sep; 173(18): 5924-5927.
doi: 10.1128/jb.173.18.5924-5927.1991

This is relevant to cite given the samples used. Did they include potential eukaryotic DNA.

7. Marine plasmids have also been discussed in :

Coastal *Synechococcus* metagenome reveals major roles for horizontal gene transfer and plasmids in population diversity
Palenik, B; Ren, Q; (...); Paulsen, IT
Feb 2009 | 11 (2) , pp.349-359

8. The early work of Sobecky is under-cited given the discussion or resistance in this paper and there are useful reviews.

Smalla, K., and Sobecky, P.A. (2002) The prevalence and diversity of mobile genetic elements in bacterial communities of different environmental habitats: insights gained from different methodological approaches. *FEMS Microbiol Ecol* 42: 165- 175.

Sobecky, P.A., and Hazen, T.H. (2009) Horizontal gene transfer and mobile genetic elements in marine systems. In *Horizontal Gene Transfer: Genomes in Flux*. M.B. Gogarten, J. Gogarten, and L. Olendzenski (eds). New York: A Humana Press, pp. 435-453.

Staff Comments:

Preparing Revision Guidelines

Please return the manuscript within 60 days; if you cannot complete the modification within this time period, please contact me. If you do not wish to modify the manuscript and prefer to submit it to another journal, please notify me of your decision immediately so that the manuscript may be formally withdrawn from consideration by Microbiology Spectrum.

Response to Reviewer comments

We would like to thank the reviewers for the useful comments and suggestions. We believe the revised version of the manuscript addresses all the comments by the reviewers. Below is our point-by-point response to the comment. The original comments are in black while our response is in red. Our response is as follows:

Reviewer #1 (Comments for the Author):

This study presents the identification of plasmids based on metagenomic analysis of published data from the Red Sea. The work is very solid and I only found a small issue that was not adequately explained:

We thank the reviewer for considering our work solid and for the suggested improvements and clarifications.

When you "merged" the output from Recycler and metaplasmidSPAdes, you found duplicates by local alignment with BLASTN. I doubt you had exactly the same result from both methods. If you didn't, what would you call "duplicates"? Sequences with >90% ID? Please clarify.

We thank the reviewer for noticing this omission and apologize for it. The cut off for merging the output from Recycler and metaplasmidSPAdes was at least 99% identity and 99% coverage. The text was edited to clarify it (lines 414-415) – "(candidates with at least 99% identity and 99% coverage)".

Another minor issue is your GitHub address. <https://github.com/Tal-Lab/Plasmidome> is not public yet?

Thank you for pointing it out. The GitHub is now public.

Other than that I enjoyed reading your work.

Thank you!

As a minor suggestion, in Fig 3 and 5, I found relatively useless panel A with the full distributions, as plasmids with many ORFs or longer than 150kb are barely seen. I'd rather keep panel B and mention in the legend that XX counts were beyond the range shown.

We thank the reviewer for this suggestion. We removed panel A from both figures and combined the two figures as panels in one figure, now numbered 4.

Reviewer #2 (Comments for the Author):

In this paper the authors analyze publicly available marine metagenome data to search for plasmids. The work develops a pipeline using two previously developed tools and a consensus approach. (Figure 1). The results are interesting but some of the putative plasmids could be chromosomal fragments etc as discussed in Antipov et al (ref 26).

We thank the reviewer for considering our work interesting and for the suggested improvements and clarifications. We are aware of the potential artifacts (both chromosomal and viral fragments) and for that reason we kept referring to our predicted plasmids as 'candidates' along the text and mentioned in the discussion (lines 399-400) that "We detected seven definite plasmids and 355 other candidate plasmids for which additional research is required to validate their plasmidic nature". Those artifacts motivated us to classify our candidates as 'plasmid', for candidates highly likely to be real plasmids, 'putative plasmid', for candidates for which there is some indication for being plasmid, and 'unknown', for candidates that we have lower confidence whether they are plasmids, either because there is no information about them or because there were some chromosomal/viral signatures. The motivation for the classification is now added to the text for clarification (lines 215-217), and the criterion for the classification appears in method section "Assignment of confidence level to plasmid candidates", lines 486-499.

Other comments:

1. I would have appreciated it if the authors did more ground-truthing of their approach, for example bioinformatically spiking in known plasmid reads (fragments) to test assembly and sensitivity to plasmid abundance.

As mentioned by Reviewer #2 above, the plasmid identification methods were previously described and were not developed as part of this study. Both methods were tested for assembly and sensitivity in the original publications (Recycler: Rozov, R. et al., *Bioinformatics* 33, 475–482 (2017); metaplasmidSPAdes: Antipov, D. et al., *Genome Res* 29, 961–968 (2019)) and in different reviews and benchmarking papers (for example, see Arredondo-Alonso, S. et al., *Microb Genom* 3, e000128 (2017); Paganini, J. A. et al., *Microorg* 9, 1613 (2021)). In this study, we do not make quantitative claims about plasmids abundance, and along the text (lines 265-270, 442-444) we clearly state that we are aware that many plasmids are missing, due to low abundance, filtering, etc.

2. The authors need to provide more information and preferably statistical analysis of the relationship of plasmids to bacteria actually present (as seem in metagenome assemblies). The authors could compare the plasmids frequencies/presence to the presence of specific taxa that can be obtained from ref 28 or related papers.

We thank the reviewer for this suggestion. An analysis of the microbial population and the correlation between plasmids and microbial presence was added (lines 142-157, 281-295, figure 3). Indeed, for some candidates, the presence of a single bacterial species can explain the presence patterns. However, this is not the case for all candidates, and there are some candidates whose pattern cannot be explained by a single bacterial species.

3. The authors need to provide more information about the samples. Specifically, the filter sizes as this is highly relevant to the interpretation of the results. " Raw Illumina sequencing paired-end reads were obtained from the NCBI BioProject database PRJNA289734 (ref 28)"

We apologize for this omittance and agree with the reviewer that it is highly important to understand the limitations of this our analysis. Thus, we added the filter sizes and their consequences to the manuscript (lines 267-268 and 409-410) – the 0.1 to 1.2 micron fraction was used.

4. The authors should provide %GC about the putative plasmids.

Average %GC content for all candidates and the %GC of the 7 plasmids were added to the main text. The %GC of all the candidates was added to Supplementary file 2.

5. Line 327 "Nevertheless, plasmid candidates that were matched to plasmids in the PLSDB were previously isolated from strains of *Vibrio parahaemolyticus*, *V. alginolyticus*, *Rhodococcus* species, and *Acinetobacter baumannii*. All these bacteria are representatives of the common marine environment phyla Proteobacteria and Actinobacteria (50)."

While the *Vibrios* are marine bacteria, *Acinetobacter* and *Rhodococcus* are soil bacteria. This statement is misleading in that the categories of Proteobacteria and Actinobacteria are too broad to be relevant. As seen in ref (19) it states that "Most replication proteins hit by the reads of SynMeta03, SynMeta04, SynMeta05 and SynMeta06 were similar to those of plasmids from several strains of *Acinetobacter baumannii*" . There is a distinct possibility that the detected plasmids are from *Synechococcus* or that there is something highly promiscuous about these plasmids. It would be best to use this information to better phrase the results.

We thank the reviewer for pointing out this misleading paragraph. The paragraph was removed and replaced with a more accurate and relevant paragraph – Lines 358-366:

"Nevertheless, plasmid candidates that were matched to plasmids in the PLSDB were previously isolated from strains of *Roseovarius* species, *Vibrio parahaemolyticus*, *V. alginolyticus*, *Rhodococcus* species, and *Acinetobacter baumannii*. *Vibrio* and *Roseovarius* genera are typical for marine environments, whereas *Rhodococcus* are mostly known as a soil bacterium. However, some *Rhodococcus* species were previously found in marine environments (54). Interestingly, though *Acinetobacter baumannii* is a soil bacterium, it was shown that the Rep proteins of many *Synechococcus* plasmids are similar to those of plasmids from several strains of *Acinetobacter baumannii* (20). Hence, it is not surprising to find matches to plasmids from *A. baumannii* in our study as well."

References

6. A reference to marine eukaryotic plasmids in Plasmids in diatom species. M Hildebrand, D K Corey, J R Ludwig, A Kukel, T Y Feng, and B E Volcani J Bacteriol. 1991 Sep; 173(18): 5924-5927.

doi: 10.1128/jb.173.18.5924-5927.1991 This is relevant to cite given the samples used. Did they include potential eukaryotic DNA.

We thank the reviewer for these relevant references. The reference and a comment about diatoms were added – “the dataset used in this study was collected from the 0.1 to 1.2 micron fraction, that excludes most of the eukaryotes, such as diatoms, which were shown to contain plasmids (44)” (lines 267-269). As the analysis was done on the 0.1 to 1.2 μm fraction, it is unlikely that diatoms and other eukaryotes are presented in the samples.

7. Marine plasmids have also been discussed in : Coastal Synechococcus metagenome reveals major roles for horizontal gene transfer and plasmids in population diversity Palenik, B; Ren, Q; (...); Paulsen, IT Feb 2009 | 11 (2) , pp.349-359

8. The early work of Sobecky is under-cited given the discussion on resistance in this paper and there are useful reviews.

Smalla, K., and Sobecky, P.A. (2002) The prevalence and diversity of mobile genetic elements in bacterial communities of different environmental habitats: insights gained from different methodological approaches. FEMS Microbiol Ecol 42: 165- 175.

Sobecky, P.A., and Hazen, T.H. (2009) Horizontal gene transfer and mobile genetic elements in marine systems. In Horizontal Gene Transfer: Genomes in Flux. M.B. Gogarten, J. Gogarten, and L. Olendzenski (eds). New York: A Humana Press, pp. 435- 453.

Thank you for your comment. The recommended references, as well as additional reference to Sobecky early work on marine plasmids, were added as references 21 (Palenik et al. 2009), 12 (Smalla & Sobecky 2002), 23 (Sobecky & Hazen 2009) and 29 (Sobecky et al. 1997).

June 13, 2023

Dr. Shay Tal
Israel Oceanographic and Limnological Research Institute
National Center for Mariculture
P.O.Box 1212
Eilat 88112
Israel

Re: Spectrum00400-23R1 (Characterization of the Environmental Plasmidome of the Red Sea)

Dear Dr. Shay Tal,

your revised manuscript has been accepted, and I am forwarding it to the ASM Journals Department for publication. You will be notified when your proofs are ready to be viewed.

Please note:

In the Legend of Fig 3 (lines 768-770), where it says both "Each row represents a sampling point" and "Each row represents a microbial genus", appears to be an error.

The first sentence should be deleted (reviewer comment).

This can be done or corrected at the editorial stage / proofs.

Please make sure to correct it in the proofs.

Sincerely,

Tino Polen
Editor, Microbiology Spectrum
